# Emerging Pharmacological Interventions for Chronic Venous Insufficiency: A Comprehensive Systematic Review and Meta-Analysis of Efficacy, Safety, and Therapeutic Advances

**DOI:** 10.3390/pharmaceutics17010059

**Published:** 2025-01-03

**Authors:** Camila Botelho Miguel, Ranielly de Souza Andrade, Laise Mazurek, Melissa Carvalho Martins-de-Abreu, Jamil Miguel-Neto, Aurélio de Melo Barbosa, Glicélia Pereira Silva, Aristóteles Góes-Neto, Siomar de Castro Soares, Javier Emilio Lazo-Chica, Wellington Francisco Rodrigues

**Affiliations:** 1Multidisciplinary Laboratory of Scientific Evidence, University Center of Mineiros (Unifimes), Mineiros 75833-130, GO, Brazil; camilabotelho@unifimes.edu.br (C.B.M.); ranielly-souza@hotmail.com (R.d.S.A.); dramelissa@unifimes.edu.br (M.C.M.-d.-A.); jamil@unifimes.edu.br (J.M.-N.); glicelia@unifimes.edu.br (G.P.S.); 2Postgraduate Program in Tropical Medicine and Infectious Diseases, Federal University of the Triângulo Mineiro—UFTM, Uberaba 38025-180, MG, Brazil; laise@unifimes.edu.br (L.M.); siomar.soares@uftm.edu.br (S.d.C.S.); 3Faculty of Physiotherapy, Universidade Estadual de Goiás—UEG, Goiânia 74015-908, GO, Brazil; aurelio.barbosa@goias.gov.br; 4Department of Microbiology, Molecular and Computational Biology of Fungi Laboratory, Instituto de Ciências Biológicas, Universidade Federal de Minas Gerais, Belo Horizonte 31270-901, MG, Brazil; arigoesneto@gmail.com; 5Laboratory of Cell Biology, Department of Structural Biology, Institute of Biological and Natural Sciences, Federal University of Triângulo Mineiro (UFTM), Uberaba 38061-500, MG, Brazil; javier.chica@uftm.edu.br

**Keywords:** chronic venous insufficiency, hydroxyethylrutoside, Pycnogenol, vasoprotective drugs, meta-analysis

## Abstract

**Background/Objectives:** Chronic Venous Insufficiency (CVI) is a progressive vascular condition characterized by venous hypertension and chronic inflammation, leading to significant clinical and socioeconomic impacts. This study aimed to evaluate the efficacy and safety of emerging pharmacological interventions for CVI, focusing on clinical outcomes such as pain, edema, cutaneous blood flow, and quality of life. **Methods:** Eligible interventions comprised new vasoprotective drugs, such as hydroxyethylrutoside, Pycnogenol, aminaphthone, coumarin + troxerutin, and Venoruton, compared to the standard therapy of diosmin and hesperidin. **Results**: Hydroxyethylrutoside and Pycnogenol showed significant benefits in pain reduction and resting flux improvement, with mean differences of 38 (95% CI: 10.56–65.44) and 25.30 (95% CI: 18.73–31.87), respectively. Improvements in edema and quality of life were less consistent. Substantial heterogeneity was observed (I^2^ = 100%, *p* < 0.001). Conclusions: Hydroxyethylrutoside and Pycnogenol emerge as promising alternatives for managing CVI. However, limitations such as high heterogeneity, small sample sizes, and methodological inconsistencies highlight the need for more robust and standardized clinical trials. This study underscores the importance of personalized and cost-effective strategies, particularly in resource-limited settings.

## 1. Introduction

Chronic Venous Insufficiency (CVI) is a progressive and multifaceted vascular disorder characterized by venous hypertension, valvular incompetence, and chronic inflammation. These pathophysiological mechanisms contribute to a spectrum of clinical manifestations, from asymptomatic varicose veins to severe complications such as venous ulcers. Symptoms including leg heaviness, pain, edema, pruritus, and nocturnal cramps are common, significantly impairing the quality of life of affected individuals. In advanced stages, CVI often leads to dermatological changes like stasis dermatitis and lipodermatosclerosis, which heighten the risk of venous ulceration and long-term morbidity [1,2].

Globally, CVI poses a significant public health challenge due to its high prevalence and socioeconomic impact. Studies estimate that approximately 25% of the population is affected by varicose veins (CEAP C2), with up to 5% developing more severe conditions, including trophic skin changes and active ulcers (CEAP C4–C6). Venous ulcers, a particularly debilitating complication, affect nearly 0.7% of the population, necessitating prolonged and costly medical care. Risk factors for CVI include female sex, increasing age, obesity, prolonged standing, and genetic predisposition. In occupational contexts, jobs requiring immobility—whether standing or sitting—are major contributors to venous hypertension, underscoring the influence of environmental and lifestyle factors on disease progression [3,4].

Pathophysiologically, CVI arises from venous reflux, obstruction, or a combination of both. These processes are exacerbated by inflammatory responses that lead to endothelial dysfunction, increased leukocyte adhesion, and vessel wall remodeling. Chronic venous hypertension, a hallmark of CVI, triggers a cascade of inflammatory mediators and adhesion molecules, contributing to structural alterations, such as increased collagen deposition and reduced elastin, ultimately culminating in skin changes and ulcer formation. Despite advances in understanding these mechanisms, the precise interplay of these factors remains incompletely elucidated, warranting further investigation [5,6].

The management of CVI involves a multidisciplinary approach encompassing conservative measures, pharmacological interventions, and minimally invasive procedures. Compression therapy, a cornerstone of treatment, is effective in alleviating symptoms and preventing disease progression by improving venous return and reducing edema.

Pharmacological therapies, particularly venotonic agents, have demonstrated efficacy in mitigating venous inflammation and enhancing microcirculatory function. Beyond the widely studied combination of diosmin and hesperidin, newer vasoprotective drugs, including hydroxyethylrutoside, Pycnogenol, aminaphthone, coumarin + troxerutin, and Venoruton, have shown promise in improving clinical outcomes, such as pain reduction, edema control, and quality of life. For instance, hydroxyethylrutoside has been reported to outperform diosmin and hesperidin in improving microvascular parameters and reducing symptoms, while Pycnogenol has demonstrated superior effects on edema and tissue oxygenation [7,8,9].

Minimally invasive techniques, such as endovenous laser ablation (EVLA) and radiofrequency ablation (RFA), have transformed the therapeutic landscape, offering comparable efficacy to traditional vein stripping but with reduced recovery times and fewer complications. However, factors such as obesity and comorbidities may influence therapeutic outcomes, highlighting the need for personalized treatment strategies. Emerging methods, including mechanochemical ablation and adhesive therapies, hold potential but require further validation through robust clinical trials [10,11,12].

Despite these advances, significant gaps remain in CVI management. Variability in therapeutic outcomes, access to advanced treatments in resource-limited settings, and inconsistencies in study methodologies hinder the development of universally applicable guidelines. Pharmacological interventions, while promising, lack extensive comparative studies to establish their relative efficacy and safety across diverse populations. Furthermore, many studies fail to standardize outcome measures, complicating meta-analyses and the generalization of findings [13,14].

This study addresses these gaps through a systematic review and meta-analysis aimed at evaluating the efficacy and safety of emerging pharmacological interventions compared to the standard diosmin-hesperidin combination. By synthesizing data on clinical outcomes such as pain, edema, cutaneous blood flow, and quality of life, this research seeks to provide robust evidence to inform clinical practice and guide the development of tailored, evidence-based treatment protocols for CVI.

## 2. Materials and Methods

### 2.1. Registration and Reporting Guidelines

This systematic review and meta-analysis was registered on the PROSPERO database (CRD42024577699), ensuring transparency and rigor in the review process. The full protocol can be accessed at The full protocol can be accessed at https://www.crd.york.ac.uk/prospero/display_record.php?ID=CRD42024577699 (accessed on 24 December 2024). The registration outlines the study’s objectives, eligibility criteria, search strategy, and planned methods for data synthesis and analysis.

The Preferred Reporting Items for Systematic Reviews and Meta-Analyses (PRISMA) 2020 guidelines were employed to guide the design, conduct, and reporting of this review. The PRISMA framework ensures adherence to established standards for systematic reviews, promoting clarity, completeness, and transparency. The methodology and results sections were developed following the PRISMA checklist and flow diagram, which are integral to this work. The updated PRISMA 2020 statement can be found in the original publication by Page et al. (2021) [15], available under a Creative Commons license (CC BY 4.0) at BMJ 2021;372.

### 2.2. Eligibility Criteria

This systematic review and meta-analysis adhered to predefined inclusion and exclusion criteria. Studies eligible for inclusion were randomized controlled trials (RCTs) involving adult participants (≥18 years) diagnosed with chronic venous insufficiency (CVI), a condition confirmed by diagnostic tools such as Doppler ultrasound. Eligible interventions comprised new vasoprotective drugs, specifically flavonoids and lactones, which were compared to the standard therapy of diosmin and hesperidin. Only studies reporting relevant clinical outcomes, such as pain (measured using validated tools like the Visual Analog Scale), edema (assessed through physical exams or objective methods such as leg circumference measurements), cutaneous blood flow (evaluated via techniques like laser Doppler flowmetry), and quality of life indices (e.g., Chronic Venous Insufficiency Questionnaire [CIVIQ]) were included.

Non-randomized studies, observational studies, case reports, and studies focusing on non-pharmacological interventions, such as surgery or compression therapy, were excluded. Similarly, studies that did not evaluate the specified outcomes or failed to compare the intervention with a valid control group were also excluded.

### 2.3. Information Sources and Search Strategy

A comprehensive search strategy was developed and applied across six major electronic databases: MEDLINE/PubMed, Cochrane Library, Embase, Scopus, Web of Science, and Google Scholar. Additionally, to ensure an inclusive capture of relevant studies, searches for grey literature were conducted on three specialized websites, including the PROSPERO database, alongside evaluations of conventional sources such as Google Scholar, MEDLINE/PubMed, and Embase for preprints and other non-conventional publications.

The search process also involved manually screening the reference lists of included studies to identify additional eligible studies. This multi-pronged approach was designed to minimize publication bias and ensure a comprehensive collection of relevant literature. The final search was completed on 25 September 2024, with no restrictions applied regarding language, publication date, or geographical location.

In total, 322 studies were retrieved from searches conducted in conventional publication sources, including MEDLINE/PubMed, Cochrane Library, Embase, Scopus, Web of Science, and Google Scholar. An additional 9 studies were identified through non-conventional sources, such as grey literature databases and preprint repositories, as well as from manual screening of reference lists of the included studies. Boolean operators (“AND”/”OR”) were used to combine keywords and phrases related to the condition, interventions, and study designs. The complete search strategies for all databases, including any filters or limits applied, are provided in Appendix A.

### 2.4. Study Selection Process

The study selection process adhered to the PRISMA 2020 guidelines. After removing 46 duplicates, the titles and abstracts of 276 unique records from conventional publication sources were screened independently by two reviewers. Of these, 178 studies were excluded due to irrelevance or failure to meet the predefined eligibility criteria. Full texts of six articles were retrieved for comprehensive assessment, resulting in the exclusion of one study for not including a comparator therapy. Ultimately, five studies from conventional sources met the eligibility criteria and were included in the qualitative synthesis.

Additionally, nine studies were identified through grey literature sources. These studies underwent full-text assessment, but none were included: seven were excluded for not including a comparator therapy, and two for not reporting the predefined outcomes.

The complete selection process, including criteria and steps for the elimination of publications, is illustrated in the PRISMA flow diagram (Figure 1). Disagreements during the selection process were resolved through consensus or consultation with a third reviewer.

### 2.5. Data Extraction and Management

Data extraction was conducted independently by two reviewers using a standardized form. Extracted data included study characteristics (e.g., authors, year, sample size, location), participant demographics (e.g., age, gender, severity of CVI), intervention details (type, dosage, duration), comparator information, and results for all specified outcomes. If necessary, study authors were contacted to clarify unclear or missing data. Discrepancies in extracted data were resolved through consensus or by involving a third reviewer. The extracted data were tabulated to ensure transparency and facilitate subsequent analyses.

### 2.6. Risk of Bias Assessment

The methodological quality of the included studies was evaluated using the Cochrane Risk of Bias 2.0 (RoB 2.0) tool. This tool assesses five key domains: the randomization process (D1), deviations from intended interventions (D2), missing outcome data (D3), measurement of outcomes (D4), and selection of reported results (D5). Each domain was classified as “low risk”, “some concerns”, or “high risk”. The overall risk of bias for each study was determined based on the following criteria: studies classified as “High risk” in at least one domain were rated as “High risk” overall; those with no “High risk” domains but with “Some concerns” in one or more domains were rated as “Some concerns” overall; and studies with “Low risk” in all domains were rated as “Low risk” overall. A traffic light plot and summary chart (Figure 2) were generated to visually represent the risk of bias across all included studies. The assessment was independently conducted by two reviewers, with discrepancies resolved through discussion or by consulting a third reviewer.

### 2.7. Data Synthesis and Statistical Analysis

All statistical analyses were performed using the R statistical software (version 4.4.2), a widely recognized platform for meta-analytical and statistical evaluations. The meta package (version 8.0-1) was employed for conducting meta-analyses, providing robust algorithms for pooling effect sizes and estimating heterogeneity [16,17,18,19,20,21,22]. The metafor package was used for meta-regression analyses to explore potential moderators [23,24], while the robvis package facilitated visualization of risk of bias assessments [25]. These tools are widely documented, validated in the literature, and commonly used in meta-analytical research, ensuring methodological rigor and reliability. Additionally, the GRADEpro GDT tool was employed to structure and visualize GRADE assessments [26], enhancing transparency and reproducibility.

To synthesize the quantitative data from the included studies, a random-effects me-ta-analysis model was employed. This model was selected due to the presence of consid-erable heterogeneity in treatment effects across studies, as indicated by metrics such as the I^2^ statistic and Cochran’s Q test. The random-effects model assumes that the true effect size may vary between studies due to differences in populations, interventions, and methodologies. The restricted maximum likelihood (REML) method was used to estimate between-study heterogeneity (τ^2^), ensuring robust modeling of variance components.

Heterogeneity was assessed using the I^2^ statistic, which quantifies the proportion of variability attributable to between-study differences rather than sampling error. The thresholds for interpretation were as follows: low (<30%), moderate (30–50%), substantial (50–75%), and considerable (>75%). Cochran’s Q test was applied to determine the statistical significance of heterogeneity, with a significance level set at *p* < 0.05. Importantly, the random-effects model was applied after identifying heterogeneity.

Mean differences (MD) with 95% confidence intervals (CIs) were used as the primary measure of effect for continuous outcomes, such as pain (measured using the Visual Analog Scale), edema (via leg circumference or volume reduction), quality of life (e.g., CIVIQ scores), and resting flux (via laser Doppler flowmetry). The use of MD allowed for the direct comparison of absolute changes between intervention and control groups. This measure was preferred over standardized mean differences (SMD) due to the consistency of measurement scales across studies, ensuring clinical interpretability and relevance.

Subgroup analyses were conducted to explore potential sources of heterogeneity, stratifying data by intervention types (e.g., hydroxyethylrutoside, aminaphthone) and clinical outcomes (e.g., pain, edema, quality of life). These analyses aimed to identify vari-ations in treatment effects attributable to specific factors. Further, meta-regression was performed using the metafor package to evaluate the influence of continuous moderators, such as publication year. The proportion of heterogeneity explained by moderators (R^2^) was reported, along with residual heterogeneity metrics (τ^2^ residual).

Forest plots were generated using the meta package to visually summarize pooled mean differences (MD) and their corresponding 95% confidence intervals (CIs), providing a clear representation of treatment effects across studies [27,28]. Funnel plots were utilized to assess the presence of publication bias, and Egger’s regression test was conducted to statistically evaluate funnel plot asymmetry [29,30,31]. Sensitivity analyses evaluated the robustness of the results by excluding individual studies to assess their influence on the pooled effect size and heterogeneity metrics. Metrics such as residual studentized values, DFFITS, and Cook’s distance were calculated to identify influential studies.

The GRADE (Grading of Recommendations, Assessment, Development, and Evalua-tions) framework was applied to evaluate the quality of evidence and the strength of rec-ommendations for each clinical outcome [26]. A bubble plot was generated using the GRADEpro GDT tool to visualize GRADE assessments, where bubble sizes represented the quality of evidence (e.g., low, moderate, high), colors indicated the strength of recommendations (e.g., strong or weak for specific interventions), and error bars denoted 95% CIs around the mean differences. This integrative approach highlighted areas of uncertainty and the reliability of the observed effects.

### 2.8. Ethical Considerations

This study was a secondary analysis of previously published data, and as such, it did not involve direct interaction with human participants or animals. Consequently, it was exempt from institutional review board approval. Nonetheless, the study adhered to the ethical principles outlined in the Declaration of Helsinki and followed all applicable guidelines and regulations to ensure integrity and ethical compliance in research. All data used in this review were obtained from publicly available sources, and no copyright or proprietary restrictions were breached during the study.

## 3. Results

### 3.1. Study Selection and Eligibility Assessment: A Comprehensive Screening Process

Initially, a total of 322 articles were identified as potentially relevant from conventional publication sources, as shown in the PRISMA flow diagram (Figure 1). These sources included MEDLINE/PubMed, Cochrane Library, Scopus, Embase, Google Scholar, and Web of Science, yielding 41, 30, 36, 36, 132, and 47 records, respectively. Of these, 46 duplicates were removed, leaving 276 unique records. After title and abstract screening, 92 studies were deemed ineligible due to not meeting predefined criteria, and 178 were excluded for irrelevance. Subsequently, six full-text articles were thoroughly reviewed, of which five met the eligibility criteria and were included in the narrative synthesis.

Additionally, nine studies were identified through searches in grey literature sources. However, none were included: seven were excluded for not including a comparator therapy, and two for not reporting the predefined outcomes.

This stepwise process of elimination is detailed in Figure 1 to provide a comprehensive overview. The characteristics of the eligible studies included in the systematic review are summarized in Table 1.

### 3.2. Risk of Bias Assessment: Methodological Evaluation and Summary Across Included Studies

The evaluation of the risk of bias in the included studies was conducted using the RoB 2.0 tool, and the findings are illustrated in Figure 2, which comprises two complementary panels: Figure 2A,B.

The individual-level risk of bias for the five included studies is detailed in Figure 2A. Each study was assessed across five methodological domains (D1 to D5), with an overall risk of bias judgment (Overall) derived using predefined criteria. The traffic light plot visually represents the results, with colors indicating the level of bias: green for “Low risk”, yellow for “Some concerns”, and red for “High risk”.

The first study exhibited “Low risk” in D1 and Overall but “High risk” in D2, D4, and D5, indicating significant concerns in multiple methodological aspects [8]. The second study demonstrated “High risk” in D1 and D4 but was classified as “Low risk” in D2, D3, and Overall [33]. For the third study [32], “Some concerns” were identified in D1, D2, and Overall, while D3 and D4 were classified as “Low risk”. The fourth study [7] revealed “Some concerns” in D1, D4, and Overall, with “Low risk” in D2 and D3. Lastly, the fifth study [34] exhibited “Some concerns” in D1, D2, and Overall, with “Low risk” in D3 and D4.

These findings highlight heterogeneity in the methodological rigor across the studies, with notable concerns in specific domains, such as deviations from intended interventions (D2) and selective outcome reporting (D5). The consistent “Low risk” ratings in D3 (missing outcome data) reflect areas of greater methodological robustness.

Figure 2B provides a summary of the proportional distribution of risk of bias judgments across all domains and the overall assessment. Each bar in the summary plot represents the proportion of studies categorized as “Low risk”, “Some concerns”, or “High risk” within a specific domain. Equal weighting was applied to all studies in the analysis. In the domain of randomization processes (D1), 40% of studies were categorized as “Low risk”, 30% as “Some concerns”, and 30% as “High risk”. For deviations from intended interventions (D2), 50% of studies were classified as “Some concerns”, 30% as “Low risk”, and 20% as “High risk”. Outcome measurement bias (D3) presented the highest proportion of “Low risk” studies (60%), with 30% showing “Some concerns” and 10% classified as “High risk”. In the domain addressing the impact of missing data (D4), 40% of studies were categorized as “Some concerns”, 30% as “Low risk”, and 30% as “High risk”. Finally, the domain concerning bias in the selection of reported results (D5) revealed 50% of studies with “Some concerns”, 30% classified as “High risk”, and 20% as “Low risk”.

In the overall judgment (Overall), 40% of studies were categorized as “Some concerns”, 30% as “Low risk”, and 30% as “High risk”. The domains with the highest frequencies of “Some concerns” or “High risk”—namely, D2 (deviations from intended interventions) and D5 (bias in the selection of reported results)—highlight critical areas for improvement in the methodological rigor of future studies. Conversely, the domain related to outcome measurement (D3) stood out for demonstrating the highest proportion of studies classified as “Low risk” (60%), reflecting greater methodological consistency in this aspect.

These findings underscore the variability in methodological quality across studies and emphasize the need for improved reporting and adherence to methodological standards, particularly in the domains of intervention deviations and the selection of reported results. The combined visualization provided in Figure 2A,B facilitates the identification of methodological strengths and weaknesses, offering a foundation for refining future research efforts.

### 3.3. Efficacy of Alternative Therapies in Reducing Chronic Venous Insufficiency Symptoms

The results presented in Figure 3A illustrate the meta-analysis of the mean differences (MD) in symptom reduction for chronic venous insufficiency (CVI) between alternative therapies and the comparator therapy (diosmin + hesperidin) across five studies. The random-effects model revealed a pooled mean difference of 19.27 (95% CI: 3.06 to 35.47, *p* = 0.0198), favoring alternative therapies. Substantial heterogeneity was observed, with a tau^2^ value of 341.34 (95% CI: 122.14 to 2823.70) and an I^2^ of 100.0%, indicating considerable variability among the included studies. The Cochran’s Q statistic (Q = 8103.22, df = 4, *p* < 0.001) further confirmed this heterogeneity. Despite this variability, the combined effect size suggests that alternative therapies provide a statistically significant benefit over diosmin + hesperidin in reducing CVI symptoms. Individual studies exhibited varying degrees of effect sizes, with MDs ranging from −2.28 (Belczak et al., 2013) [34] to 37.33 (Cesarone et al., 2005) [7].

Figure 3B extends the analysis to the stratification of all observed biological effects, including pain, edema, quality of life, and resting flux, across 15 observed effects between the studies. The random-effects model yielded a pooled mean difference (MD) of 18.79 (95% CI: 9.19 to 28.39, *p* < 0.0001), favoring alternative therapies over diosmin + hesperidin. Heterogeneity remained high, with a tau^2^ of 359.11 (95% CI: 192.01 to 897.08) and an I^2^ of 100.0%. The Cochran’s Q statistic (Q = 97,254.74, df = 14, *p* < 0.001) confirmed significant variability. These findings suggest that study-level characteristics, such as differences in intervention protocols, population demographics, and methodological approaches, may contribute to the observed heterogeneity.

The stratification of biological outcomes in Figure 3B underscores the consistent therapeutic benefits of alternative therapies across multiple dimensions of CVI symptomatology. However, the pronounced heterogeneity necessitates cautious interpretation of the pooled results and highlights the importance of subgroup or sensitivity analyses to better understand the variability.

### 3.4. Subgroup Analysis of Interventions for Chronic Venous Insufficiency: Comparative Efficacy Across Alternative Therapies

The subgroup analysis was conducted to evaluate the impact of different drugs used as interventions for reducing symptoms associated with chronic venous insufficiency (CVI). The analysis utilized a random-effects model, considering mean differences (MD) and their respective 95% confidence intervals (CIs) for each subgroup. These findings are illustrated in Figure 4.

In Figure 4A, the meta-analysis results were stratified by type of intervention, encompassing five studies. The random-effects model revealed an overall MD of 19.27 (95% CI: 3.06 to 35.47; *p* = 0.0198), indicating a statistically significant benefit of alternative therapies compared to diosmin + hesperidin. However, heterogeneity was substantial, with tau^2^ = 341.34 and I^2^ = 100.0%, reflecting high variability among studies.

Within the subgroups, hydroxyethylrutoside demonstrated the greatest effect with an MD of 37.33 (95% CI: 37.17 to 37.49), while Pycnogenol showed an MD of 14.97 (95% CI: −12.59 to 42.53). Interventions with coumarin + troxerutin or aminaphthone and Venoruton yielded MDs of −2.28 (95% CI: −3.44 to −1.11) and 31.30 (95% CI: 29.70 to 32.90), respectively. Heterogeneity within subgroups was also significant, particularly for Pycnogenol (tau^2^ = 394.46; I^2^ = 99.8%). The test for subgroup differences confirmed significant variability among the interventions (Q = 4402.75; *p* < 0.001).

Figure 4B presents an adjusted analysis in which coumarin + troxerutin and aminaphthone, initially grouped in the study by Belczak et al. (2013) [34], were analyzed separately. This adjustment revealed an overall MD of 15.68 (95% CI: 0.69 to 30.67; *p* = 0.0404), indicating a beneficial effect of the interventions. Heterogeneity remained high (tau^2^ = 350.47; I^2^ = 99.9%).

Within this analysis, hydroxyethylrutoside continued to demonstrate the most significant benefit (MD = 37.33; 95% CI: 37.17 to 37.49), followed by Pycnogenol (MD = 14.97; 95% CI: −12.59 to 42.53). Conversely, coumarin + troxerutin and aminaphthone exhibited negative effects, with MDs of −2.89 (95% CI: −4.76 to −1.02) and −1.67 (95% CI: −3.15 to −0.18), respectively. The test for subgroup differences remained statistically significant (Q = 4391.30; *p* < 0.001), confirming notable variation among the interventions.

Figure 4C extends the subgroup analysis by incorporating various biological outcomes, including pain, edema, quality of life, and resting flux. This comprehensive analysis included 15 studies and revealed an overall MD of 18.79 (95% CI: 9.19 to 28.39; *p* = 0.0001), underscoring the consistent benefits of alternative therapies. However, heterogeneity remained exceptionally high (tau^2^ = 359.11; I^2^ = 100.0%).

Among the subgroups, hydroxyethylrutoside exhibited the most pronounced benefit (MD = 37.33; 95% CI: 22.80 to 51.86; tau^2^ = 164.81; I^2^ = 100.0%), followed by Pycnogenol (MD = 21.04; 95% CI: 9.29 to 32.80; tau^2^ = 250.77; I^2^ = 99.9%). In contrast, aminaphthone and coumarin + troxerutin displayed MDs of −1.58 (95% CI: −20.66 to 17.50) and −2.66 (95% CI: −8.74 to 3.42), respectively. The test for subgroup differences remained statistically significant (Q = 124.98; *p* < 0.0001), reinforcing the variability in effects among the interventions.

### 3.5. Subgroup Analysis of Clinical Manifestations in Chronic Venous Insufficiency: Pain, Edema, Quality of Life, and Resting Flux

The subgroup analysis stratified by different clinical manifestations of chronic venous insufficiency (CVI), including pain, edema, quality of life, and resting flux, is presented. Figure 5 illustrates the results of this analysis, conducted using a random-effects model, reporting mean differences (MD) with their respective 95% confidence intervals (CIs). The overall meta-analysis, encompassing 15 evaluations of the different clinical manifestations derived from the five eligible studies, revealed a pooled MD of 18.79 (95% CI: 9.19 to 28.39; *p* = 0.0001), favoring other therapies over diosmin + hesperidin. However, heterogeneity across studies was extremely high, with tau^2^ = 359.11 and I^2^ = 100.0%, indicating substantial variability among the included evaluations.

The analysis for pain included two evaluations, yielding a pooled MD of 38.00 (95% CI: −139.89 to 215.89), which indicates a significant mean difference favoring other therapies. However, heterogeneity within this subgroup was exceptionally high (tau^2^ = 391.84; I^2^ = 100.0%), likely attributable to differences in patient populations and intervention protocols. For edema, seven evaluations were analyzed, resulting in a pooled MD of 12.85 (95% CI: −3.36 to 29.05). Although the effect was positive, the confidence interval included the null value, indicating a lack of statistical significance. Heterogeneity was again substantial in this subgroup (tau^2^ = 476.41; I^2^ = 99.8%). When assessing quality of life, three evaluations were included, revealing a pooled MD of 13.20 (95% CI: −5.08 to 31.48). Similar to edema, the confidence interval crossed zero, suggesting no significant difference. Heterogeneity remained high for this outcome (tau^2^ = 260.32; I^2^ = 99.7%). For resting flux, three evaluations demonstrated a statistically significant pooled MD of 25.30 (95% CI: 18.73 to 31.87), strongly favoring other therapies. While heterogeneity was relatively lower compared to other subgroups, it was still considerable (tau^2^ = 33.67; I^2^ = 100.0%).

The test for subgroup differences did not reveal statistically significant variability between the manifestations (Q = 4.23; *p* = 0.238), indicating that the observed effects were relatively consistent across the evaluated manifestations, despite differences in magnitude. The high degree of heterogeneity observed across all subgroups underscores the influence of study-level characteristics, such as variations in intervention types, clinical populations, and outcome measurement methods. These findings emphasize the necessity for further investigation to understand the sources of variability and to confirm the observed benefits of other therapies over diosmin + hesperidin for managing CVI symptoms.

### 3.6. Temporal Trends in Intervention Effectiveness: Meta-Regression Analysis

The results of a mixed-effects meta-regression model evaluating the temporal effects of publication year on the mean differences in clinical outcomes across 15 assessments related to chronic venous insufficiency (CVI) interventions are presented in Figure 6. This analysis incorporates the year of publication as a moderator to investigate its influence on the variability of effects.

The model revealed substantial residual heterogeneity, with tau^2^ estimated at 95.47 (SE = 37.85), reflecting considerable variability in effect sizes among studies even after accounting for the publication year. The I^2^ value was 100.0%, indicating that all residual variability originated from between-study heterogeneity rather than sampling error. Furthermore, the H^2^ value of 27,991.19 emphasized the pronounced variability. Notably, the R^2^ value was 73.41%, demonstrating that the year of publication accounted for a significant proportion of the observed heterogeneity, underscoring its importance as a moderator.

The test for residual heterogeneity (QE = 27,259.01; *p* < 0.0001) confirmed significant variability among studies, even after adjusting for the publication year. Similarly, the test for the moderator effect (QM = 39.03; *p* < 0.0001) demonstrated that the year of publication was a statistically significant moderator, effectively explaining a substantial portion of the observed heterogeneity.

The regression model yielded an intercept of 7640.69 (SE = 1219.96; *p* < 0.0001), representing the mean effect size when the year of publication is zero—a purely statistical reference for the regression line. The regression coefficient for the year variable was −3.79 (SE = 0.61; *p* < 0.0001), with a 95% confidence interval ranging from −4.98 to −2.60. This negative coefficient indicates a significant downward trend in effect sizes over time, suggesting that the effectiveness of interventions has decreased by approximately 3.79 units per year.

These findings highlight a significant temporal trend, where effect sizes have declined in more recent studies. This decline could be attributed to evolving clinical practices, methodological improvements, or changes in the perceived efficacy of interventions. Despite the year of publication accounting for a substantial proportion of variability (R^2^ = 73.41%), a significant amount of residual heterogeneity remains, indicating that other factors beyond the publication year influence effect sizes.

In summary, the meta-regression underscores the critical role of the year of publication as a significant moderator, revealing a notable temporal dimension to the variability in intervention effectiveness for CVI. However, the persistence of high residual heterogeneity suggests the need for further research to identify additional moderators and fully elucidate the sources of variability in outcomes across studies.

### 3.7. Assessment of Publication Bias and Heterogeneity in Alternative Therapies for Chronic Venous Insufficiency

The evaluation of publication bias was conducted using funnel plots and Egger’s regression test for small-study effects. Figure 7A presents the analysis restricted to the studies included in the primary meta-analysis (k = 5), while Figure 7B extends the evaluation to include all observed biological effects (k = 15), such as pain, edema, quality of life, and resting flux.

For Figure 7A, the funnel plot illustrates the distribution of effect sizes against their corresponding standard errors for the five studies included in the primary analysis. The Egger’s regression test yielded a t-value of −1.48 (df = 3) and a *p*-value of 0.2362, suggesting no statistically significant evidence of asymmetry in the funnel plot (i.e., no small-study effects). The bias estimate was −37.89 (SE = 25.65), with a tau^2^ value of 1563.94, indicating substantial heterogeneity among the studies. Despite the lack of significant asymmetry, the limited number of studies (k = 5) reduces the statistical power of Egger’s test, warranting cautious interpretation of these results.

Figure 7B includes all studies stratified by biological effect, assessing a broader range of clinical outcomes. The funnel plot shows a more extensive distribution of effect sizes. The Egger’s regression test for this dataset produced a t-value of 0.69 (df = 13) and a *p*-value of 0.5044, further indicating no significant evidence of publication bias or small-study effects. The bias estimate was 17.46 (SE = 25.43), and the tau^2^ value was substantially higher at 7219.35, reflecting increased heterogeneity due to the inclusion of diverse clinical outcomes and study designs. Notably, the observed heterogeneity may also be attributable to the different types of interventions categorized under “other therapies”, such as hydroxyethylrutoside, Pycnogenol, aminaphthone, coumarin + troxerutin, and Venoruton, all compared against diosmin + hesperidin.

The overall distribution of studies in both funnel plots appeared symmetrical around the pooled mean effect size, supporting the absence of strong publication bias. However, the high heterogeneity observed in both analyses (tau^2^ = 1563.94 and 7219.35 for Figure 7A and Figure 7B, respectively) highlights variability among studies.

### 3.8. Evaluation of Evidence Quality and Recommendation Strength Across Clinical Outcomes Using Grade Assessment

The results of the GRADE assessment are presented in a bubble plot (Figure 8), offering a comprehensive visualization of the mean differences in effect sizes across the evaluated clinical outcomes—pain, edema, quality of life, and resting flux. The x-axis represents the outcomes, while the y-axis illustrates the mean differences in effect sizes. Bubble sizes correspond to the quality of evidence, categorized as “Low” (small bubbles), “Moderate” (medium bubbles), and “High” (large bubbles). Bubble colors denote the strength and direction of the recommendation: blue for “Strong for Other Therapies”, red for “Weak for Other Therapies”, green for “Strong for Diosmin + Hesperidin”, and orange for “Weak for Diosmin + Hesperidin”. Error bars on each bubble indicate the 95% confidence intervals (CIs), with overlapping intervals signaling uncertainty in some outcomes.

For the outcome “Pain”, a significant mean difference of +38 was observed, supported by moderate-quality evidence. This result was accompanied by a strong recommendation for other therapies (blue bubble), reflecting consistent and meaningful benefits in managing pain relative to diosmin + hesperidin. Despite the positive findings, the moderate evidence level suggests that additional high-quality studies are necessary to reinforce these conclusions.

The outcome “Edema” demonstrated a mean difference of +12.85, with CIs spanning from −3.36 to 29.05, indicating a lack of statistical significance. The overlapping CIs underscore the uncertainty in the effect size, resulting in a weak recommendation for other therapies (red bubble). The moderate quality of evidence highlights the need for further investigation to clarify the potential benefits of these therapies in reducing edema.

For “Quality of Life”, the mean difference was +13.2, supported by moderate-quality evidence. However, the CIs ranged from −5.09 to 31.48, including the null value, which diminishes the confidence in the observed benefit. Consequently, a weak recommendation for other therapies (red bubble) was issued, signaling the necessity of robust clinical trials to confirm these findings.

The outcome “Resting Flux” exhibited a mean difference of +25.3, indicating substantial benefits of other therapies. Despite the low-quality evidence associated with this outcome (small bubble), the strong recommendation for other therapies (blue bubble) reflects a compelling signal of efficacy. This result is particularly noteworthy given concerns about potential publication bias, emphasizing the need for cautious interpretation and further validation.

In summary, the GRADE assessment highlights varying degrees of evidence quality and recommendation strength across clinical outcomes. While strong recommendations favoring other therapies were issued for pain and resting flux, the weak recommendations for edema and quality of life underscore the need for more rigorous and definitive studies to address the observed uncertainties. These findings provide critical insights into the effectiveness of alternative therapies in managing chronic venous insufficiency (CVI) symptoms, while also emphasizing the importance of evidence-based decision-making in clinical practice.

### 3.9. Sensitivity Analyses by Intervention Types and Clinical Manifestations

The sensitivity analyses aimed to evaluate the robustness of the observed effects by stratifying the studies based on intervention types and clinical manifestations. These analyses, detailed in Appendix A, explored whether subgroups could explain the variability in the observed effects, providing additional insights into the factors influencing the outcomes.

The mixed-effects model for sensitivity analysis by intervention type (Appendix A) included 13 different outcomes for clinical manifestations (studies) and revealed substantial residual heterogeneity, with tau^2^ estimated at 190.70 (SE = 95.91) and an I^2^ of 100.0%, indicating that all residual variability was attributable to between-study heterogeneity rather than sampling error. The R^2^ value of 24.10% suggested that intervention types accounted for 24.1% of the observed heterogeneity. The residual heterogeneity test (QE = 13,464.31, *p* < 0.0001) confirmed significant variability among the studies, even after adjusting for the subgroups. The moderator test (QM = 7.78, *p* = 0.0999) indicated a near-significant trend, suggesting potential differences between intervention types, although the results did not reach statistical significance at the 5% level. Within the subgroups, hydroxyethylrutoside and Venoruton demonstrated positive trends in effect size (21.93 and 23.23, respectively), while coumarin + troxerutin showed a negative trend (−10.94). However, none of these subgroup effects were statistically significant, with *p*-values exceeding 0.05.

The sensitivity analysis stratified by clinical manifestations (Appendix A) similarly demonstrated persistent high heterogeneity, with tau^2^ estimated at 306.11 (SE = 144.85) and an I^2^ of 100.0%. The R^2^ value was 0.00%, indicating that clinical manifestations such as pain, quality of life, and resting flux did not account for any of the variability between studies. The residual heterogeneity test (QE = 77,381.76, *p* < 0.0001) highlighted significant variability, and the moderator test (QM = 0.88, *p* = 0.8313) confirmed no significant differences between subgroups based on clinical manifestations. Among the subgroups, pain demonstrated a positive mean difference (7.09), as did resting flux (8.39), while quality of life showed a negative trend (−3.72). However, none of these subgroup effects were statistically significant, and confidence intervals overlapped zero, indicating substantial uncertainty in the estimates.

These findings emphasize the persistence of substantial heterogeneity across studies, which remains largely unexplained by either intervention type or clinical manifestation. While intervention types accounted for a modest portion of the variability (R^2^ = 24.10%), clinical manifestations did not contribute to explaining heterogeneity (R^2^ = 0.00%). The lack of significant subgroup differences in both analyses underscores the complexity of the factors driving variability and highlights the need for further investigation into other potential moderators influencing the observed effects.

### 3.10. Influence Analysis of Clinical Evaluations and Interventions in the Meta-Analysis

The influence analysis was conducted to assess the impact of individual evaluations on the overall meta-analytic results, with the findings summarized in Appendix A. This analysis provides valuable insights into whether certain evaluations disproportionately influence the pooled effect size or observed heterogeneity, offering a deeper understanding of the stability of the meta-analysis.

The residual studentized values (rstudent) revealed that the evaluation by Cesarone et al. (2005), manifestation: pain, intervention: hydroxyethylrutoside had the highest value (rstudent = 1.9986), approaching the critical threshold of ±2. This suggests that this evaluation may be an outlier or exert significant influence on the meta-analytic results. Other evaluations exhibited rstudent values closer to zero, indicating a lower degree of deviation from the overall effect.

The DFFITS metric, which measures the impact of excluding a specific evaluation on the pooled effect size, highlighted the evaluations by Cesarone et al. (2005) [7], manifestation: pain, intervention: hydroxyethylrutoside (DFFITS = 0.5324) and Belczak et al. (2013) [34], manifestation: edema, intervention: coumarin + troxerutin (DFFITS = −0.4700) as having relatively higher influence compared to the remaining evaluations. These findings suggest that removing these evaluations could alter the estimated effect size more substantially than others.

Cook’s Distance (Cook.D), a composite measure of influence, corroborated these observations. The evaluation by Cesarone et al. (2005) [7], manifestation: pain, intervention: hydroxyethylrutoside had the largest Cook.D value (0.2333), identifying it as the most influential evaluation in the meta-analysis. In contrast, evaluations such as Cesarone et al. (2006) [8], manifestation: edema, intervention: Pycnogenol (Cook.D = 0.0059) and Cesarone et al. (2006) [33], manifestation: quality of life, intervention: Venoruton (Cook.D = 0.0011) exhibited negligible influence, suggesting that their exclusion would have minimal impact on the pooled estimates.

Regarding the variance among evaluations (tau^2^.del), excluding the evaluation by Cesarone et al. (2005) [7], manifestation: pain, intervention: hydroxyethylrutoside substantially reduced the estimated heterogeneity (tau^2^.del = 295.5462), compared to the initial tau^2^ value (tau^2^ = 7219.3471). This indicates that this evaluation significantly contributes to the observed heterogeneity. Similarly, the heterogeneity test statistic (QE.del) demonstrated a marked decrease when this evaluation was removed (QE.del = 91,366.9214), suggesting its inclusion inflates the observed heterogeneity across evaluations. Conversely, the exclusion of evaluations such as Toledo et al. (2017) [32], manifestation: edema, intervention: Pycnogenol and Belczak et al. (2013) [34], manifestation: quality of life, intervention: aminaphthone had minimal impact on both tau^2^.del and QE.del, reinforcing their low influence on the overall model.

These results indicate that the evaluation by Cesarone et al. (2005) [7], manifestation: pain, intervention: hydroxyethylrutoside is the most influential, consistently showing high values across multiple metrics, including rstudent, DFFITS, Cook.D, and pronounced effects on tau^2^.del and QE.del. This suggests that this evaluation may be an outlier or possess unique methodological characteristics that differentiate it from others. Further investigation into its design, sample size, methodology, and quality is warranted to understand its substantial impact.

Additionally, the evaluation by Belczak et al. (2013) [34], manifestation: edema, intervention: coumarin + troxerutin also exhibited moderate influence, as evidenced by its DFFITS and Cook.D values. While its impact is less pronounced than that of Cesarone et al. (2005) [7], manifestation: pain, it remains notable within the context of the meta-analysis.

These findings highlight the importance of influence analysis in identifying evaluations that disproportionately affect the pooled effect size and heterogeneity. This information aids in interpreting the robustness of the meta-analytic conclusions and emphasizes the need for careful scrutiny of the most influential evaluations to ensure the reliability of the results.

## 4. Discussion

The findings of this study underscore the multifaceted nature of chronic venous insufficiency (CVI) and the range of therapeutic approaches required for its management. CVI remains a multifactorial challenge, necessitating the integration of pharmacological, minimally invasive, and conservative therapies tailored to individual patient profiles. A comprehensive review of the literature provides an insightful perspective on the effectiveness, limitations, and implications of these strategies. The integration of these approaches emerges as a cornerstone of effective CVI care, synthesizing evidence on their efficacy and limitations while providing critical insights for advancing clinical practice and research.

Flavonoids stand out as a key pharmacological intervention due to their venotonic, antioxidative, and anti-inflammatory properties. Among these, the combination of diosmin and hesperidin (D + H) has been extensively studied. Cesarone et al. (2005) demonstrated the superiority of hydroxyethylrutoside (HR) over D + H in improving microcirculatory parameters, such as resting skin flux (RF) and capillary filtration rate (RAS), while significantly reducing symptom scores. These outcomes suggest that HR not only restores venous function but also contributes to enhanced patient quality of life. Similarly, Pycnogenol has shown remarkable efficacy [7]. Cesarone et al. (2006) highlighted its ability to outperform D + H in reducing edema, improving tissue oxygenation, and alleviating venous symptoms [8]. These findings are consistent with those of Casili et al. (2021), who underscored the therapeutic potential of flavonoids such as troxerutin, diosmin, and horse chestnut extract in modulating venous tone and reducing endothelial inflammation [9]. However, Rodrigues et al. (2024) pointed to the heterogeneity in study designs and methodologies, emphasizing the need for more rigorous and standardized clinical trials to validate these outcomes [35].

Minimally invasive techniques, including endovenous laser ablation (EVLA) and radiofrequency ablation (RFA), have revolutionized CVI management. Studies by Gao et al. (2022) and Bellmunt-Montoya et al. (2021) demonstrated that these approaches are as effective as traditional vein stripping, with lower complication rates and faster recovery times [11,12]. However, the CHIVA method offers a unique perspective by prioritizing the preservation of venous anatomy. While CHIVA showed lower recanalization rates compared to conventional techniques, it was associated with a higher incidence of bruising [12]. Despite this limitation, its conservative approach may benefit patients seeking anatomical preservation. Furthermore, Deol et al. (2020) observed that increased body mass index (BMI) diminishes the effectiveness of minimally invasive techniques like thermal ablation, emphasizing the need for personalized treatment strategies that incorporate weight management [10].

Venous ulcers, a severe complication of CVI, represent a significant challenge in clinical practice. Toledo et al. (2017) demonstrated that both Pycnogenol and D + H effectively reduced ulcer diameters and limb circumferences, supporting their role as adjuvant therapies [32]. Cesarone et al. (2006) reinforced these findings, highlighting Pycnogenol’s superior outcomes in clinical and microcirculatory parameters, such as edema reduction and tissue perfusion enhancement [8]. These results align with the broader benefits of venotonic therapies on patient quality of life. Cesarone et al. (2006) and Belczak et al. (2014) reported significant improvements in health-related quality of life scores, particularly with aminaphtone, which demonstrated notable efficacy in alleviating venous symptoms [8,34]. Chaitidis et al. (2022) emphasized the complementary role of pharmacological treatments when combined with compressive and procedural interventions [36].

Conservative measures remain foundational in the management of chronic venous insufficiency, with elastic compression playing a central role in alleviating symptoms and reducing venous pressure. Complementary approaches, such as physical activity programs designed to strengthen the calf muscle pump and improve ankle joint mobility, have shown potential in mitigating symptoms of chronic venous insufficiency. Araujo et al. (2016) conducted a systematic review highlighting the benefits of exercise interventions in improving venous return and reducing edema. Despite methodological limitations, including small sample sizes and a high risk of bias, the review reported significant improvements in parameters such as venous refilling time and muscle strength at faster speeds among intervention groups [37].

In older patients, rehabilitation strategies are particularly important due to the high prevalence of polypharmacy and its associated risks. Giovannini et al. (2020) observed that antidepressant use among older adults in long-term care facilities is influenced by various patient and institutional factors, which can affect clinical outcomes in the management of chronic venous insufficiency. Integrating physical activity programs tailored for this population could mitigate the adverse effects of immobility and drug interactions while enhancing treatment efficacy and overall quality of life [38].

Complementary strategies, such as balneotherapy, have also shown promise. De Moraes Silva et al. (2023) reviewed evidence suggesting that balneotherapy can improve clinical severity and quality of life, although limitations in study design warrant cautious interpretation [39]. During the COVID-19 pandemic, venotonic agents proved critical in reducing venous stasis and complications associated with restricted mobility [40]. These findings underscore the enduring relevance of conservative therapies in diverse clinical contexts.

The socio-economic burden of CVI is particularly pronounced in low-resource settings. Disparities in access to advanced therapies, as highlighted by Azar et al. (2022) and Javier and Ortiz (2020), necessitate scalable, cost-effective solutions [13,14]. While EVLA and RFA are recommended as first-line treatments in industrialized nations, limited availability in resource-constrained regions underscores the importance of public health investments in infrastructure and research [11]. The integration of cost-effective alternatives, such as CHIVA or conservative therapies, may bridge this gap and expand access to care.

Despite these advancements, several limitations persist in CVI management. Methodological inconsistencies across studies, including variability in outcome measures and the absence of blinding, challenge the generalizability of findings. Jantet (2002) emphasized the heterogeneity of CVI presentations, influenced by demographic and comorbid factors [41]. Krasinski et al. (2021) noted that the pandemic exacerbated challenges in CVI care, further emphasizing the need for adaptive strategies [40]. Orhurhu et al. (2021) advocated for comprehensive pain management approaches, integrating pharmacological and procedural therapies to address the multidimensional impact of CVI [42].

Future research should prioritize standardizing clinical outcomes, conducting multicentric trials, and exploring factors such as genetic predispositions, regional disparities, and comorbid conditions. Chaitidis et al. (2022) proposed multicentric studies to validate innovative approaches, such as CHIVA, balneotherapy, and advanced flavonoid therapies, across diverse populations and stages of CVI [36]. These efforts are crucial for refining treatment paradigms and optimizing patient care.

The integration of personalized, evidence-based strategies combining pharmacological advancements, minimally invasive techniques, and conservative therapies holds immense potential to improve the management of chronic venous insufficiency (CVI). These approaches offer a multidimensional framework to optimize clinical outcomes, reduce symptom burden, and enhance the quality of life for patients with this prevalent condition. By addressing existing gaps in research and implementation, there is an opportunity to refine treatment paradigms, ensuring that care is not only effective but also equitable across diverse populations and healthcare settings.

However, several limitations should be considered when interpreting the findings of this study. First, the heterogeneity of the included studies poses challenges to synthesizing the evidence and drawing generalizable conclusions. Variability in study designs, outcome measures, and methodologies often complicates comparisons and may introduce bias into the analysis. For example, differences in the selection of study populations, the definition of clinical endpoints, and the duration of follow-ups are significant sources of variability.

Additionally, the reliance on secondary data introduces potential constraints related to publication bias and the quality of the original studies. While rigorous methods were employed to assess the risk of bias, including the use of standardized tools like ROB2, methodological weaknesses in some studies, such as the absence of blinding and inconsistent reporting, may have influenced the findings. These limitations highlight the need for more robust, multicenter randomized clinical trials to validate the efficacy and safety of innovative interventions across diverse populations and settings.

Another limitation lies in the scope of therapies evaluated. Although this review provides a comprehensive assessment of pharmacological, procedural, and conservative treatments, it does not address the long-term sustainability and cost-effectiveness of these strategies, particularly in low-resource settings. The socio-economic barriers to accessing advanced therapies, such as endovenous ablation techniques, remain an important consideration that warrants further exploration.

Lastly, the findings are constrained by the lack of standardized reporting of comorbid conditions, such as obesity, diabetes, and cardiovascular diseases, which are known to influence treatment outcomes in CVI. A more detailed stratification of results based on these moderating factors could provide deeper insights into personalized treatment approaches.

However, while the integration of evidence-based pharmacological advancements, minimally invasive techniques, and conservative therapies represents a promising strategy to optimize CVI management, these findings should be interpreted with caution due to the inherent limitations of the data. Addressing these gaps through more rigorous and standardized research will not only strengthen the evidence base but also pave the way for the development of tailored interventions that meet the diverse needs of CVI patients. By overcoming these challenges, it will be possible to enhance care quality, mitigate disease burden, and ensure equitable access to effective treatments for all affected individuals.

## 5. Conclusions

This systematic review and meta-analysis provides a robust evaluation of emerging pharmacological interventions for Chronic Venous Insufficiency (CVI). Among the therapies assessed, hydroxyethylrutoside and Pycnogenol demonstrated the most consistent and significant benefits, including improvements in pain reduction, edema control, and microvascular function, compared to the standard diosmin-hesperidin combination. These findings suggest their potential as effective alternatives in managing CVI symptoms.

Conversely, aminaphthone and coumarin + troxerutin did not show significant benefits or demonstrated negative trends in some analyses, highlighting the need for further high-quality studies to better understand their clinical utility. The overall heterogeneity observed across studies underscores the variability in therapeutic responses and emphasizes the importance of standardized protocols and consistent outcome measures in future research.

While the results highlight the promise of specific pharmacological interventions, they also reveal significant gaps in evidence, particularly for certain therapies. Future research should address these limitations by conducting rigorous trials with larger, diverse populations to validate these findings and inform tailored, evidence-based treatment strategies for CVI.

## Figures and Tables

**Figure 1 pharmaceutics-17-00059-f001:**
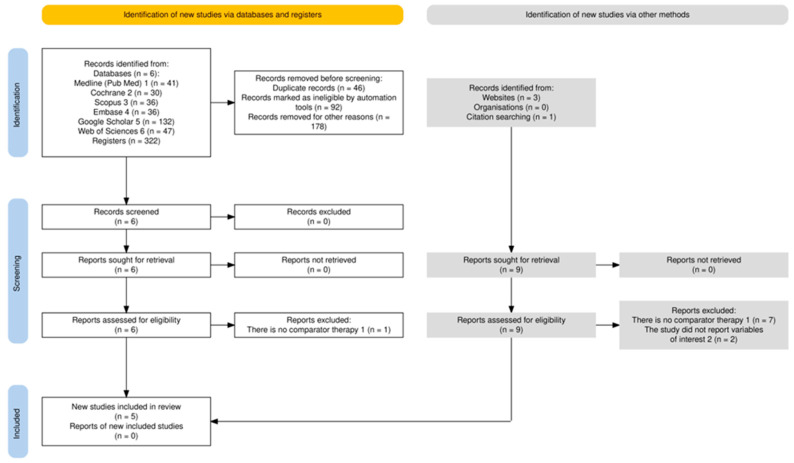
Flow diagram summarizing the study selection process. A total of 322 articles were identified, of which 46 duplicates were removed, 92 were excluded for ineligibility, and 178 were discarded for other reasons. Six full-text articles were reviewed, and five met the eligibility criteria. Additional searches retrieved four more records, but nine were excluded after eligibility assessment (PRISMA 2020) [15].

**Figure 2 pharmaceutics-17-00059-f002:**
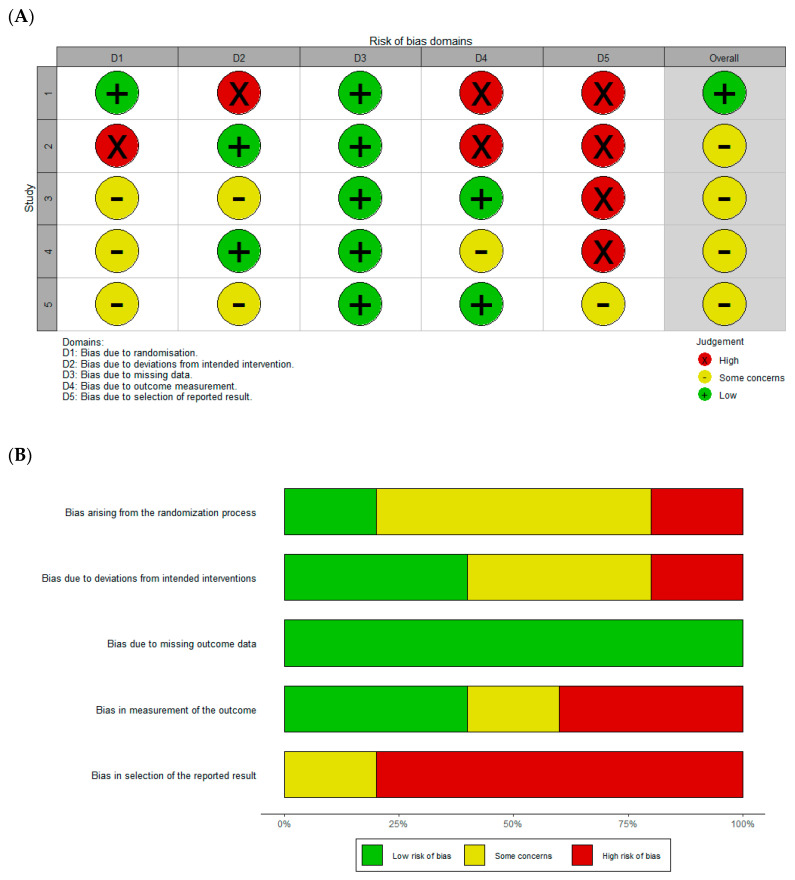
Risk of bias assessment using ROB2 tool. (**A**) Traffic light plot illustrating the risk of bias judgments for each domain (D1 to D5) and the overall assessment (Overall) for each included study. Green indicates “Low risk”, yellow represents “Some concerns”, and red denotes “High risk”. The studies evaluated are as follows: Cesarone et al. (2006) [8], Cesarone et al. (2006) [33], Toledo et al. (2017) [32], Cesarone et al. (2005) [7], and Belczak et al. (2013) [34]. (**B**) Bar chart summarizing the proportions of risk of bias across all methodological domains (D1 to D5) and the overall assessment (Overall) for all included studies. Each bar reflects the distribution of risk levels, categorized as “Low risk”, “Some concerns”, and “High risk”. This plot highlights the areas of methodological concern, such as deviations from intended interventions (D2) and selective outcome reporting (D5), while identifying domains with greater methodological consistency, such as missing outcome data (D3).

**Figure 3 pharmaceutics-17-00059-f003:**
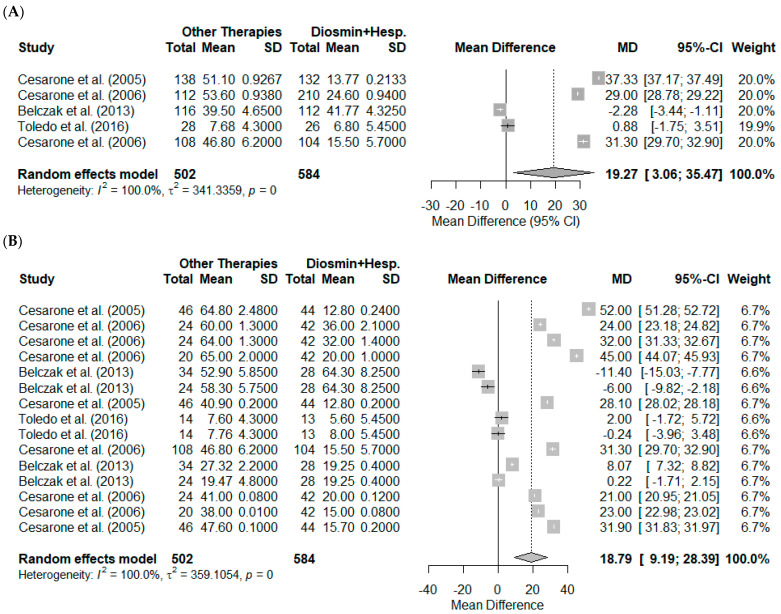
Meta-analysis results comparing alternative therapies with diosmin + hesperidin for the treatment of chronic venous insufficiency (CVI). (**A**) Pooled mean differences (MD) in symptom reduction across five studies using a random-effects model, highlighting variability in effect sizes [7,8,32,33,34]. (**B**) Stratification of biological effects, including pain, edema, quality of life, and resting flux, across 15 studies, showing significant overall benefits of alternative therapies. Error bars represent 95% confidence intervals (CI), and heterogeneity metrics (I^2^ and tau^2^) indicate substantial variability across studies [7,8,32,33,34].

**Figure 4 pharmaceutics-17-00059-f004:**
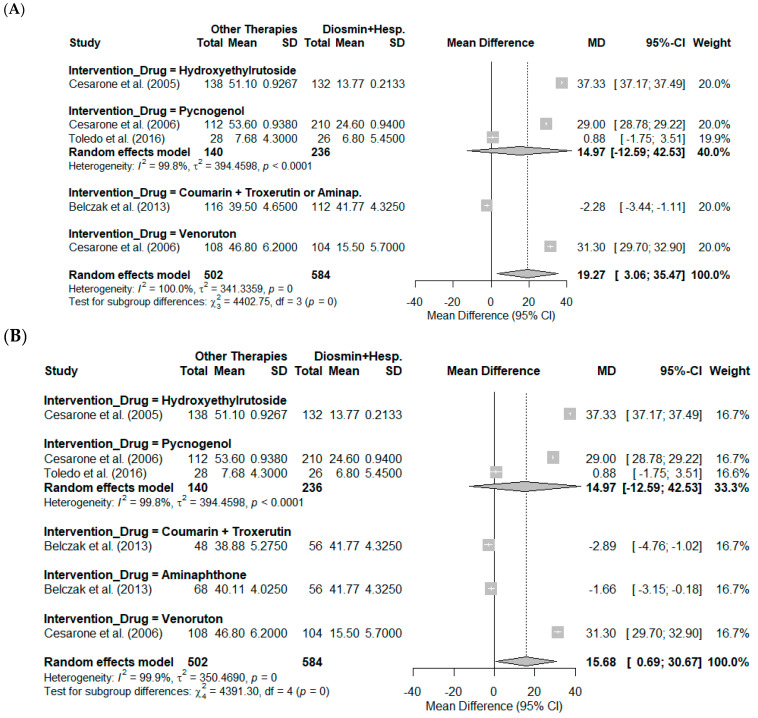
Subgroup analysis of mean differences (MD) in symptom reduction for chronic venous insufficiency (CVI) comparing alternative therapies to diosmin + hesperidin. (**A**) displays the analysis with interventions grouped as reported in the original studies (k = 5), demonstrating significant variability across interventions [7,8,32,33,34]. (**B**) separates interventions initially grouped under coumarin + troxerutin or aminaphthone (k = 6), revealing distinct effects for each [7,8,32,33,34]. (**C**) stratifies results by biological outcomes, including pain, edema, quality of life, and resting flux (k = 15). Subgroups include hydroxyethylrutoside, Pycnogenol, aminaphthone, coumarin + troxerutin, and Venoruton. The overall pooled effects, subgroup estimates, and heterogeneity metrics (I^2^ and tau^2^) are presented for each analysis [7,8,32,33,34].

**Figure 5 pharmaceutics-17-00059-f005:**
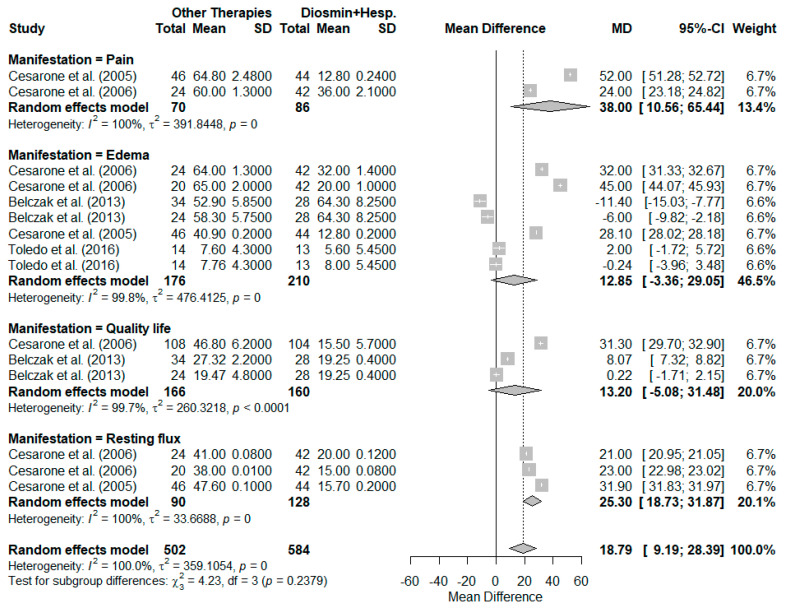
Subgroup analysis of the mean differences (MD) in symptom reduction for chronic venous insufficiency (CVI), stratified by clinical manifestations: pain, edema, quality of life, and resting flux. The pooled MDs and their 95% confidence intervals (CIs) are shown for each subgroup using a random-effects model. The overall analysis included 15 assessments across five eligible studies, highlighting significant benefits for certain manifestations, particularly resting flux and pain. Heterogeneity was substantial across all subgroups, with tau^2^ and I^2^ values indicating variability in study-level characteristics and methodologies [7,8,32,33,34].

**Figure 6 pharmaceutics-17-00059-f006:**
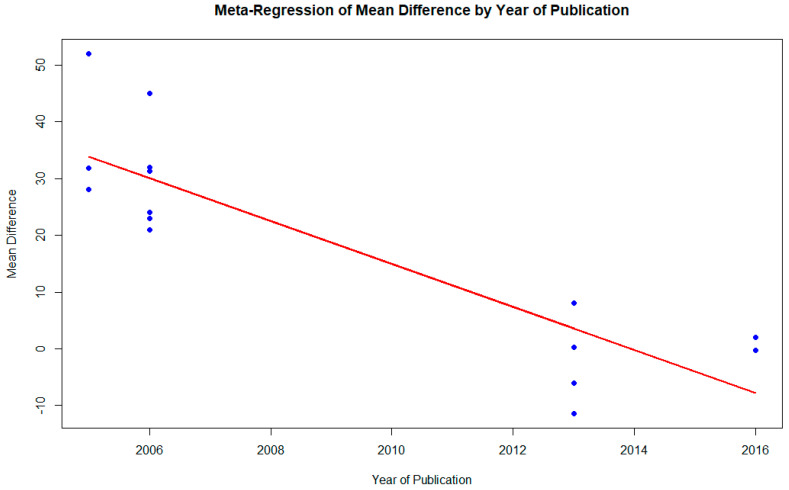
Results of the mixed-effects meta-regression model assessing the impact of publication year on mean differences in clinical outcomes for chronic venous insufficiency (CVI) interventions across 15 assessments. The analysis revealed a significant negative temporal trend, indicating a decrease in effect sizes over time. Residual heterogeneity was substantial (tau^2^ = 95.47; I^2^ = 100.0%), with the year of publication accounting for 73.41% of the variability (R^2^ = 73.41%). The regression coefficient for the year variable was −3.79 (95% CI: −4.98 to −2.60; *p* < 0.0001), reflecting a significant decline in intervention effectiveness over the years.

**Figure 7 pharmaceutics-17-00059-f007:**
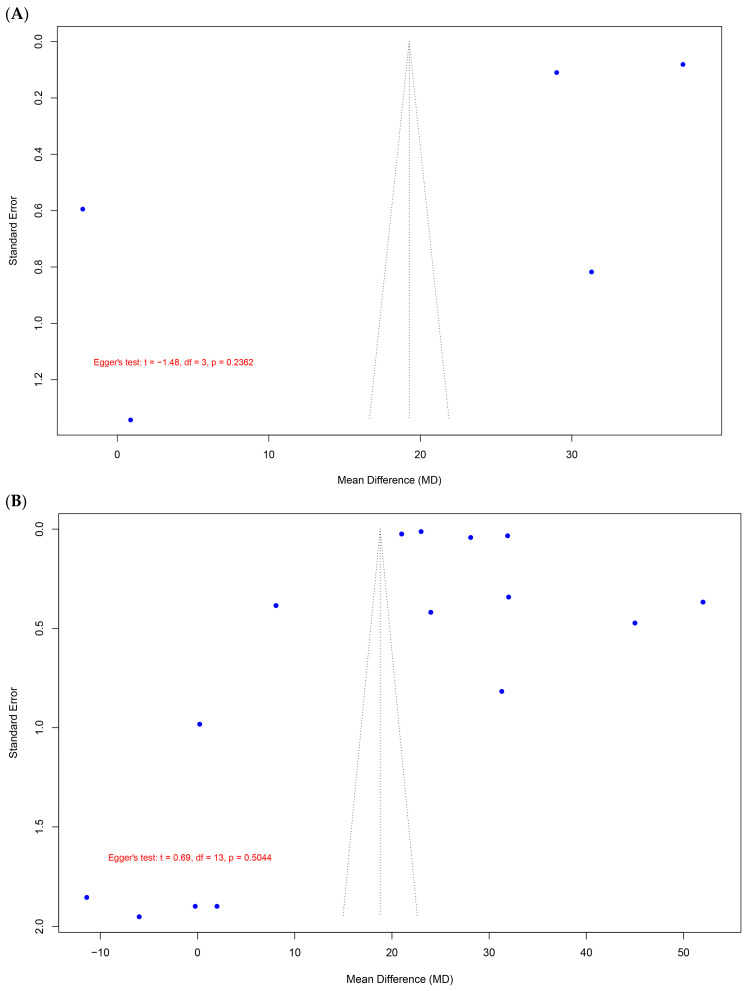
Funnel plot analysis of publication bias for studies evaluating alternative therapies for chronic venous insufficiency. (**A**) Funnel plot representing the primary meta-analysis results (k = 5). Egger’s regression test indicated no significant asymmetry (t = −1.48, df = 3, *p* = 0.2362), with a bias estimate of −37.89 (SE = 25.65) and tau^2^ = 1563.94. (**B**) Funnel plot encompassing all stratified biological effects, including pain, edema, quality of life, and resting flux (k = 15). Egger’s regression test showed no significant asymmetry (t = 0.69, df = 13, *p* = 0.5044), with a bias estimate of 17.46 (SE = 25.43) and tau^2^ = 7219.35. Both analyses suggest the absence of strong publication bias, although high heterogeneity reflects variability in study methodologies, populations, and the different types of interventions categorized as “Other therapies”, including hydroxyethylrutoside, Pycnogenol, aminaphthone, coumarin + troxerutin, and Venoruton, compared to diosmin + hesperidin.

**Figure 8 pharmaceutics-17-00059-f008:**
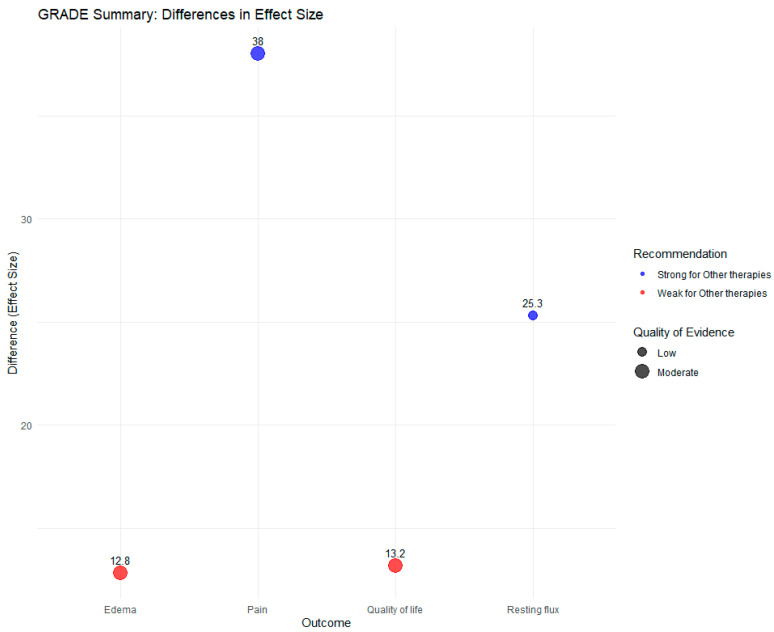
Bubble plot summarizing the GRADE assessment of mean differences in effect sizes across clinical outcomes in chronic venous insufficiency (CVI). The x-axis represents the outcomes (pain, edema, quality of life, and resting flux), while the y-axis indicates the mean differences (effect sizes). Bubble sizes reflect the quality of evidence: “Low” (small), “Moderate” (medium), and “High” (large). Bubble colors indicate the strength of recommendations: blue for “Strong for Other Therapies”, red for “Weak for Other Therapies”, green for “Strong for Diosmin + Hesperidin”, and orange for “Weak for Diosmin + Hesperidin”. Numeric values within the plot denote the mean differences for each outcome, and error bars represent 95% confidence intervals. Overlapping intervals highlight uncertainties in specific outcomes.

**Table 1 pharmaceutics-17-00059-t001:** Characteristics of eligible studies in the systematic review.

Study (Year)	Intervention	Interventions Effects	Main Findings
Cesarone et al., (2006) [8]	Pycnogenol 150 e 300 mg	Pycnogenol 150 mg/day significantly improved all microcirculatory parameters after four weeks (*p* < 0.05), with twice the reduction in edema, rest flow, and ankle swelling compared to Daflon. Pycnogenol 300 mg/day was more effective in reducing edema, but other parameters showed no clear dose-dependent effect. Daflon 1000 mg/day showed minor, non-significant improvements in most parameters and slight increases in pO_2_ and pCO_2_ compared to Pycnogenol.	Pycnogenol demonstrated superior efficacy compared to Daflon in improving signs, symptoms, and microcirculatory parameters in patients with chronic venous insufficiency (CVI) and venous microangiopathy. It resulted in greater reductions in edema, resting flux, and ankle swelling rate. The 300 mg dose of Pycnogenol was generally more effective than the 150 mg dose in reducing edema.
Toledo et al., (2017) [32]	Pycnogenol 150 mg	Pycnogenol and diosmin/hesperidin significantly reduced ulcer area over time (*p* < 0.001). The Pycnogenol group showed a trend toward better ulcer healing during the 90-day study and achieved a significant reduction in ankle circumference earlier (45 days) compared to the diosmin/hesperidin group (60 days).	Treatment with Pycnogenol and diosmin/hesperidin significantly reduced the circumference of the proximal and distal thirds of the ankle over time. Pycnogenol showed a trend toward a more favorable effect on ulcer healing compared to diosmin/hesperidin. No significant differences were observed between the groups in color Doppler ultrasound results.
Cesarone et al., (2006) [33]	Oxerutins	A 46.8% reduction in Ve-QOL score was observed in the Pycnogenol group after 8 weeks (*p* < 0.05), compared to a 15.5% reduction in the diosmin/hesperidin group. The 31.3% difference in score variation between the groups was statistically significant (*p* < 0.05).	Treatment with oxerutins (Venoruton) significantly improved the venous quality of life score (Ve-QOL) by 46.8%, compared to a 15.5% improvement with D + H (Daflon). Oxerutins outperformed D + H in alleviating signs and symptoms of chronic venous insufficiency (CVI), including venous microangiopathy and edema. The treatment was well tolerated, with no dropouts due to side-effects or clinical concerns.
Belczak et al., (2013) [34]	Aminaphthone//Coumarin + Troxerutin	Limb volume reduction: 64.3% of the diosmin + hesperidin group, 58.3% of the coumarin + troxerutin group, 52.9% of the aminaphtone group, and 36.6% of the placebo group achieved a volume reduction ≥100 mL. Quality of life (QOL) improvement: The aminaphtone group showed a 27.4% improvement in QOL scores, followed by the diosmin + hesperidin group (19.2%), the coumarin + troxerutin group (18.1%), and the placebo group (9.2%). Specific symptom improvements included significant relief of edema and pain/burning in the aminaphtone and diosmin + hesperidin groups, significant improvement in heaviness/fatigue in the aminaphtone group, and significant improvement in pruritus/paresthesia in the coumarin + troxerutin group.	No differences were observed in tibio-tarsal joint range of motion between treatment groups. Venoactive drugs (VAD), particularly aminaphtone and diosmin + hesperidin, outperformed placebo in improving quality of life (QoL). VADs, especially aminaphtone, diosmin + hesperidin, and coumarin + troxerutin, effectively alleviated specific symptoms of chronic venous disease, including edema, pain/burning, heaviness/fatigue, and itching.
Cesarone et al., (2005) [7]	HR (Venoruton)	Resting skin flow (RF) decreased by 47.6% in the HR group and 15.7% in the D + H group. Ankle swelling rate (RAS) dropped by 40.9% in the HR group and 12.8% in the D + H group. Analog Symptom Line Score (ASLS) was reduced by 64.8% in the HR group and 12.9% in the D + H group. All changes were statistically significant (*p* < 0.05) in the HR group, while those in the D + H group were smaller and not significant.	Oral treatment with HR (0-[beta-hydroxyethyl]-rutosides) demonstrated rapid and significant effectiveness in improving microcirculation in patients with chronic venous insufficiency (CVI) and venous microangiopathy. The HR group showed notable reductions in resting skin flux and capillary filtration, accompanied by significant improvements in CVI signs and symptoms. HR outperformed the diosmin and hesperidin (D + H) combination in enhancing microcirculatory parameters and alleviating CVI-related signs and symptoms.x1

## Data Availability

The data supporting the findings of this study are included in the article and its Appendix A. Additionally, the research protocol is publicly accessible on the PROSPERO database under registration number CRD42024577699 and can be found at https://www.crd.york.ac.uk/prospero/display_record.php?ID=CRD42024577699 (accessed on 24 December 2024). Further inquiries can be directed to the corresponding author.

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
