# Peer review of "Emerging Pharmacological Interventions for Chronic Venous Insufficiency: A Comprehensive Systematic Review and Meta-Analysis of Efficacy, Safety, and Therapeutic Advances"

_pharmaceutics, 2025, doi:10.3390/pharmaceutics17010059_

Round 1

Reviewer 1 Report

Comments and Suggestions for Authors

hydroxyethylrutoside, pycnogenol, aminaphtone, coumarin + troxerutin, and Venoruton, comparing them with standard therapy, in which diosmin and hesperidin are used. The effectiveness of the pharmacological action of the above-mentioned drugs in eliminating pain, swelling, blood flow through the skin, and also quality of life was taken into account. They showed that the most promising therapy is the use of hydroxyethylrutoside and pycnogenol, which effectively in reducing pain and improving resting flow. However, they were less effective in improving swelling and quality of life. Both drugs appear to be suitable for the treatment of chronic venous insufficiency. In turn, the therapy with the combination of drugs: aminaphthone and coumarin + troxerutin did not provide significant benefits and sometimes showed negative effects in some analyses. The Authors believe that this requires further high-quality studies to better orient their usefulness in CVI therapy. However, preliminary data require further more robust and standardized clinical studies.

The work is interesting, deepening our knowledge about CVI therapy. Thanks to the use of advanced analytical/statistical methods, such as the meta-Analysis registered in Prospero at the University of York, UK using the PRISMA method with the use of other tools such as Cochrane Risk of Bias 2.0. 322 articles were selected by elimination according to various criteria, only 5 were selected.

The Authors demonstrated the superiority and greater benefits of therapy conducted with hydroxyethylrutoside and pycnogenol than the traditional therapy with a combination of diosmin and hesperidin.

-I think it would be worth emphasizing clearly that the process of eliminating publications is presented in the scheme Fig. 1

-It would be good to clearly state the number of publications selected in relation to the different pharmacological strategies assessed

-It is necessary to provide the statistical package that was used for statistical analysis of the data.

-If necessary, state how it was validated? Were the R scripts written by the authors and if so, were they somehow verified to validate them

-Clearly explain what measure of effect was used in the meta-analysis

-It would be advisable to clearly explain why there were 9 publications and 5 included studies.

-It would be helpful to explain to the readers and clearly state how the overall bias was estimated for the 5 publications (Fig. 2a)

Table 1 Oxirutins correct Oxirutins

Line 124 The Authors write "Additionally, studies that measured monocyte chemoattractant protein-1 (MCP-1) as an inflammatory marker were considered for inclusion".

However, I did not find any results in the text devoted to this marker.

Author Response

The work is interesting, deepening our knowledge about CVI therapy. Thanks to the use of advanced analytical/statistical methods, such as the meta-Analysis registered in Prospero at the University of York, UK using the PRISMA method with the use of other tools such as Cochrane Risk of Bias 2.0. 322 articles were selected by elimination according to various criteria, only 5 were selected.

The Authors demonstrated the superiority and greater benefits of therapy conducted with hydroxyethylrutoside and pycnogenol than the traditional therapy with a combination of diosmin and hesperidin.

-I think it would be worth emphasizing clearly that the process of eliminating publications is presented in the scheme Fig. 1

Answer: We sincerely thank you for your careful review and dedication to improving the quality of our manuscript. Your insightful comments have significantly contributed to making the text clearer and more robust.

Regarding your suggestion to emphasize the process of eliminating publications in Figure 1, we fully agree with your observation. To address this, we have made revisions in two specific sections of the manuscript:

"Study Selection Process" Section (Methodology):

We revised the text to explicitly highlight that the elimination process is represented in the PRISMA flow diagram (Figure 1). The updated sentence is as follows:

Revised Text:

"The complete selection process, including the criteria and steps for the elimination of publications, is clearly illustrated in the PRISMA flow diagram (Figure 1)."

Section 3.1: Study Selection and Eligibility Assessment (Results):

In this section, we clarified the stepwise elimination process and emphasized its connection to Figure 1. The revised passage is as follows:

Revised Text:

"Initially, a total of 322 articles were identified as potentially relevant, as shown in the PRISMA flow diagram (Figure 1). [...] This stepwise process of elimination is detailed in Figure 1 to provide a comprehensive overview."

All changes are highlighted in yellow in the new version of the manuscript for your convenience. We hope these updates address your concerns and provide greater clarity regarding the elimination process. Please let us know if further refinements are needed.-It would be good to clearly state the number of publications selected in relation to the different pharmacological strategies assessed

-It is necessary to provide the statistical package that was used for statistical analysis of the data.

Answer: Thank you for your observation regarding the need to specify the statistical package used in the analysis. We appreciate your attention to detail and have revised the manuscript to provide greater clarity on this point. The information regarding the statistical software and packages employed has been highlighted in the "Data Synthesis and Statistical Analysis" section. Specifically, we have detailed the use of R statistical software (version 4.4.2) along with key packages, such as meta for meta-analysis, metafor for meta-regression, and robvis for visualizing risk of bias. Additionally, the GRADEpro GDT tool was used for structuring the GRADE assessments. These updates are highlighted in yellow in the revised manuscript for your convenience.

-If necessary, state how it was validated? Were the R scripts written by the authors and if so, were they somehow verified to validate them

Answer: Thank you for raising this important point regarding the validation of the R scripts used in our analyses. To clarify: The scripts utilized in this study were not written by the authors but were directly extracted from the functions provided in the R packages meta, metafor, and robvis. These packages are well-documented, extensively validated in the literature, and widely used for meta-analytical and statistical analyses.

To enhance reproducibility and transparency, we have included appropriate citations for each package and tool used in the revised manuscript. These references provide detailed information about the functionality, algorithms, and validation processes of the tools applied.

We hope this clarification addresses your concerns and have highlighted these updates in yellow in the revised manuscript for your convenience.

-Clearly explain what measure of effect was used in the meta-analysis

Answer: The primary measure of effect for the meta-analysis was the mean difference (MD) with 95% confidence intervals (CIs). The MD was chosen due to the consistency of measurement scales across the included studies, allowing for direct and clinically interpretable comparisons between intervention and control groups. Outcomes analyzed included pain (e.g., via Visual Analog Scale), edema (e.g., leg circumference), quality of life (e.g., CIVIQ scores), and resting flux (e.g., laser Doppler flowmetry).

-It would be advisable to clearly explain why there were 9 publications and 5 included studies.

Answer: Thank you for your thorough review and for bringing this issue to our attention. Your observation allowed us to identify and address an inconsistency in the description of our methodology and results.

Upon careful review of the manuscript, we clarified that six studies were reviewed in full from the conventional databases, not nine as previously stated. The nine studies referenced were identified through searches in grey literature sources; however, none of these were ultimately included in the analysis. Specifically, seven studies from the grey literature were excluded for not including a comparator therapy, and two were excluded for not reporting the predefined outcomes of interest.

To address this, we have revised the Methods and Results sections to accurately reflect this process. We also expanded the description of the grey literature search to provide additional detail and ensure transparency. These changes include specifying the sources used for grey literature searches, the criteria applied for their evaluation, and the reasons for their exclusion. Furthermore, the PRISMA flow diagram and accompanying narrative have been adjusted to align with these revisions, offering a comprehensive overview of the study selection process.

To assist your review, all changes made to the manuscript in response to this comment have been highlighted in yellow. We believe these modifications significantly improve the clarity and consistency of the manuscript, aligning the text with the actual methodology followed during the study.

We sincerely appreciate your valuable feedback, which has strengthened the overall quality of our manuscript. If any further clarification or additional details are required, we are happy to provide them.

-It would be helpful to explain to the readers and clearly state how the overall bias was estimated for the 5 publications (Fig. 2a)

Answer: Thank you for your observation and for highlighting the need to provide greater clarity regarding the overall risk of bias estimation for the five included studies.

The methodological quality of the included studies was assessed using the Cochrane Risk of Bias 2.0 (RoB 2.0) tool, as described in the Methods section. This tool evaluates five key domains: the randomization process (D1), deviations from intended interventions (D2), missing outcome data (D3), measurement of outcomes (D4), and the selection of reported results (D5). For each domain, studies were classified as having "Low risk," "Some concerns," or "High risk."

To estimate the overall risk of bias for each study, the judgments for all five domains were summarized based on the guidance provided by the RoB 2.0 tool. Specifically:

If a study was classified as "High risk" in at least one domain, the overall risk of bias was also deemed "High risk."

If no domains were classified as "High risk" but one or more domains raised "Some concerns," the overall risk of bias was classified as "Some concerns."

If all domains were classified as "Low risk," the overall risk of bias was judged as "Low risk."

These criteria were consistently applied across all five included studies, and the results are illustrated in Figure 2a. The traffic light plot provides a visual representation of the individual and overall risk of bias assessments. Additionally, discrepancies during the evaluation process were resolved through discussion or consultation with a third reviewer to ensure accuracy and consistency in the judgments.

We have revised the Results section to provide a more explicit explanation of how the overall risk of bias was determined, ensuring that readers can clearly understand the process. The revised text includes the specific criteria for summarizing domain-level judgments into an overall bias judgment. Furthermore, we added a note in the figure legend of Figure 2 to reinforce this explanation.

These modifications are intended to enhance transparency and clarity in the reporting of the risk of bias assessment. The updated text has been highlighted in yellow for ease of identification. We hope this addresses your concern, and we remain available to provide further clarification if needed.

Table 1 Oxirutins correct Oxirutins

Answer: We sincerely thank the reviewer for their careful observation regarding the inconsistency in Table 1, where "Oxirutinas" was incorrectly mentioned. We have corrected this to "Oxerutins" and ensured that the term is consistently used throughout the manuscript. Additionally, we conducted a thorough review of the entire manuscript to identify and address any other potential errors. All necessary corrections have been made and are clearly marked in the revised document for ease of review.

Line 124 The Authors write "Additionally, studies that measured monocyte chemoattractant protein-1 (MCP-1) as an inflammatory marker were considered for inclusion". However, I did not find any results in the text devoted to this marker.

Answer: Thank you for your careful review and for pointing out the mention of MCP-1 as an inclusion criterion in line 124. Initially, we included MCP-1 as a potential marker due to its relevance in inflammatory processes associated with chronic venous insufficiency. However, upon reviewing the eligible studies, we found that none reported data on MCP-1, which limited its applicability within the scope of our review.

To avoid inconsistencies and ensure clarity for readers, we have decided to remove this criterion from the text. We greatly appreciate your observation, as it allowed us to enhance the overall coherence and accuracy of the manuscript.

Reviewer 2 Report

Comments and Suggestions for Authors

The authors presented a systematic review and meta-analysis about pharmacological interventions for chronic venous insufficiency. In order to assess the safety and effectiveness of new pharmacological treatments for chronic venous insufficiency, the authors compared new vasoprotective medications to the conventional diosmin and hesperidin therapy, paying particular attention to clinical outcomes like pain, edema, cutaneous blood flow, and quality of life.

Abstract: clear and well written.

Introduction: Good

Materials and Methods: Well written and clear. Methods are correct and robust.

Results: clear and well expressed

Discussion and conclusion: good, including limitations.

Just a minor concern: Even though pharmacological treatments appear promising, and considering—as said by the authors—the need for more robust evidence, more study is still required, especially to determine the best treatment plans and long-term effects for different patient populations. In the discussion, please consider the possible role of rehabilitation: Treatments like physical activity that strengthen the muscle pump in the leg's calf and increase ankle joint mobility may help lessen the symptoms (10.1002/14651858.CD010637.pub2). This is particularly important in older patients where polypharmacy and the use of several drugs, such as antidepressants, could impact clinical practice (10.1186/s12877-020-01730-5).

Author Response

The authors presented a systematic review and meta-analysis about pharmacological interventions for chronic venous insufficiency. In order to assess the safety and effectiveness of new pharmacological treatments for chronic venous insufficiency, the authors compared new vasoprotective medications to the conventional diosmin and hesperidin therapy, paying particular attention to clinical outcomes like pain, edema, cutaneous blood flow, and quality of life. Abstract: clear and well written. Introduction: Good Materials and Methods: Well written and clear. Methods are correct and robust. Results: clear and well expressed. Discussion and conclusion: good, including limitations. Just a minor concern: Even though pharmacological treatments appear promising and considering—as said by the authors—the need for more robust evidence, more study is still required, especially to determine the best treatment plans and long-term effects for different patient populations. In the discussion, please consider the possible role of rehabilitation: Treatments like physical activity that strengthen the muscle pump in the leg's calf and increase ankle joint mobility may help lessen the symptoms (10.1002/14651858.CD010637.pub2). This is particularly important in older patients where polypharmacy and the use of several drugs, such as antidepressants, could impact clinical practice (10.1186/s12877-020-01730-5).

Answer: Thank you for your kind and positive feedback regarding our manuscript. We are delighted that you found the abstract, introduction, materials and methods, results, discussion, and conclusions to be well-written and robust. Your acknowledgment of the clarity and strength of our work is greatly appreciated.

We also value your insightful suggestion about incorporating the role of rehabilitation, particularly physical activity interventions aimed at strengthening the calf muscle pump and improving ankle joint mobility. This is indeed an important consideration, especially in older populations, where polypharmacy and the use of medications such as antidepressants may significantly influence clinical practice.

Based on your recommendation, we have expanded the discussion section to address the potential benefits of physical activity and rehabilitation in chronic venous insufficiency management, referencing the articles you provided (10.1002/14651858.CD010637.pub2 and 10.1186/s12877-020-01730-5). These additions enhance the comprehensiveness of our discussion by highlighting complementary approaches that can improve clinical outcomes and patient quality of life.

We sincerely appreciate your constructive feedback, which has helped us further strengthen our manuscript.
